# Identification of Risk Factors in Patients with Recurrent Cystitis May Improve Individualized Management

**DOI:** 10.3390/diagnostics15222885

**Published:** 2025-11-14

**Authors:** Jakhongir F. Alidjanov, Ulugbek A. Khudaybergenov, Khurshid B. Khudayberdiev, Jennifer Kranz, Laila Schneidewind, Fabian P. Stangl, José Medina-Polo, Adrian Pilatz, Tommaso Cai, Kurt G. Naber, Florian M. Wagenlehner, Truls E. Bjerklund Johansen

**Affiliations:** 1Scientific Team of the Acute Cystitis Symptom Score, Medical Biostatistics and Data Science, 90449 Nuremberg, Germany; 2Department of Urology, Tashkent State Medical University, Tashkent 100100, Uzbekistan; 3Department of Urology and Pediatric Urology, University Clinic of Aachen, 52074 Aachen, Germany; 4Department of Urology and Kidney Transplantation, Martin-Luther-University, 06097 Halle, Germany; 5Department of Urology, Inselspital, University of Bern, 3010 Bern, Switzerland; 6Department of Urology, Faculty of Medicine, Medical Center-University of Freiburg, 79106 Freiburg, Germany; 7Department of Urology and Andrology, Salzburg University Hospital, Paracelsus Medical University, 5020 Salzburg, Austria; 8Urology Department, Hospital Universitario 12 de Octubre Imas12, 28041 Madrid, Spain; 9Department of Urology, Andrology and Pediatric Urology, Justus-Liebig University of Giessen, 35392 Giessen, Germanyflorian.wagenlehner@chiru.med.uni-giessen.de (F.M.W.); 10Interdepartmental Centre of Medical Sciences (CISMed), University of Trento, 38122 Trento, Italy; ktommy@libero.it; 11Urology Division (APSS), Provincial Health Care Agency, Santa Chiara Regional and Teaching Hospital, 38122 Trento, Italy; 12Department of Urology, Technical University of Munich, 81675 Munich, Germany; 13Department of Urology, Oslo University Hospital, Institute of Clinical Medicine, University of Oslo, 0318 Oslo, Norway; 14Institute of Clinical Medicine, University of Aarhus, 8000 Aarhus, Denmark

**Keywords:** urinary tract infections, acute cystitis, recurrent cystitis, risk factors, antimicrobial stewardship

## Abstract

**Background/Objectives**: Management of acute episodes of lower urinary tract infection (LUTI) depends on whether they are sporadic or recurrent. We aimed to define factors that differentiate patients with acute sporadic cystitis (AC) from those with recurrent cystitis (RC) and thereby improve individualized care. **Methods**: We performed a post hoc analysis of prospectively collected data from the multinational GPIU.COM study. Female patients with an acute LUTI episode completed the Acute Cystitis Symptom Score (ACSS) and underwent a routine clinical and laboratory evaluation, including a physical examination, ultrasonography, urinalysis, and urine culture and antimicrobial susceptibility testing. Risk factors for recurrence were evaluated using the Lower Urinary Tract Infection Recurrence Risk (LUTIRE) nomogram and the ORENUC classification. Statistical analysis followed a robust stepwise approach. Significant variables were assessed by relative risk (RR), and logistic regression was used to estimate odds ratios (ORs). Model performance was evaluated using the area under the curve (AUC), the Hosmer–Lemeshow test, variance inflation factor (VIF), and bootstrap sampling. **Results**: A total of 106 women were included (AC *n* = 50; RC *n* = 56). Patients with RC more frequently presented with a history of constipation, a severe impact of symptoms on daily activities, multiple uropathogens, and trace proteinuria. Pyuria was inversely associated with RC. Logistic regression identified chronic constipation, severe impact of symptoms on daily activities, and multiple uropathogens as independent predictors of RC. Three predictive models showed consistent discrimination between AC and RC (AUC = 0.80, 0.82, and 0.84). **Conclusions**: AC and RC showed notable differences in certain symptom profiles, quality of life, urinalysis, and microbiological findings. Combining high-value predictors from LUTIRE and ORENUC into a comprehensive prognostic algorithm could improve assessment of recurrence risk. A refined classification of LUTIs with recurrence grading is warranted to guide decision-making and prevention strategies.

## 1. Introduction

Acute episodes of lower urinary tract infections (LUTIs) are among the most prevalent bacterial infections in women and typically present with dysuria, frequency, and urgency [1]. LUTIs can present as both acute sporadic (AC) and recurrent cystitis (RC). RC is defined as at least two episodes within six months or three episodes within one year, and it may persist for decades, impairing patients’ quality of life by affecting social and sexual relationships, self-esteem, and work productivity and increasing the healthcare burden on society [2,3].

Patients with RC receive numerous courses of antibiotics, which puts them at risk of developing resistant pathogens that require treatment with broad-spectrum antibiotics with unwanted collateral damage [4]. To counteract this trend and adhere to antibiotic stewardship principles, the medical community has developed new strategies against LUTIs, such as avoiding antibiotic treatment by using non-antibiotic therapy only (e.g., NSAIDs), delayed antimicrobial therapy, and replacing empiric broad-spectrum antibiotics with tailored treatment based on urine culture and susceptibility testing [5,6,7,8]. To prevent recurrences, patients might be offered prophylactic measures with nutraceuticals, antiadhesives, hormone replacement, or antibiotics, although some interventions (e.g., D-mannose) have recently failed to show efficacy in randomized controlled trials [9]. Today, treatment and diagnostic evaluation of LUTIs are differentiated depending on whether the acute cystitis is sporadic or recurrent [1].

A key feature in the diagnostic work-up and the prevention of recurrences is the identification of factors that predispose to recurrent infection, increase the risk of a more severe outcome, and demand a longer treatment period [10,11,12,13]. Various classifications of risk factors have been suggested to ensure a comprehensive patient evaluation in the search for risk factors that can be eliminated (LUTIRE nomogram; ORENUC classification) [11,12]. Hitherto, evaluation of risk factors has mainly been discussed in patients with severe upper-tract UTI, but there is also a spectrum of risk factors for patients with less severe lower-tract UTI. Recent reviews highlight the role of host immune mechanisms and genetic susceptibility in recurrent UTIs, which may reflect differences in innate response mechanisms between patients with AC and RC [14,15,16,17].

The present analytical study aimed to improve individualized management of cystitis by identifying patients at risk of recurrence, refining diagnostic evaluation, and informing tailored preventive measures. The prioritized objective was to determine the factors that distinguish AC from RC.

## 2. Materials and Methods

### 2.1. Ethical Considerations

The primary study protocol was approved by the Ethics Committee of the Justus-Liebig University of Giessen, Germany (AZ.:10/15, 4 August 2015) [12]. Written informed consent was obtained from all respondents prior to enrollment. Personal data were anonymized in accordance with applicable data protection regulations [18].

### 2.2. Study Design and Definitions

The current work is a post hoc analysis of prospectively collected data within the multinational, prospective non-interventional Global Prevalence Study of Infections in Urinary Tract in Community Setting (GPIU.COM) investigating the epidemiology of acute LUTI in community-based settings [19].

The diagnosis of an acute episode of LUTI was made at the patient’s initial visit based on symptoms, relevant medical history, and a positive clean-catch midstream urine dipstick test showing ≥25 white blood cells (WBC)/µL by leucocyte esterase. The urine culture was used to identify the causative pathogen and determine its antimicrobial susceptibility profile.

Recurrence was defined as two or more acute episodes of LUTI within the preceding six months or three or more acute episodes within the preceding 12 months [1].

### 2.3. Study Population, History and Physical Examination

Women aged 16 years and older presenting with a suspected acute episode of LUTI were recruited from outpatient clinics in Germany and Switzerland between July 2016 and February 2023.

At study entry, each respondent completed a paper study report form (SRF) covering demographics, medical history, and medications. Height, weight, and body mass index (BMI) were recorded, and all patients underwent routine clinical examination and abdominal ultrasonography by a urologist.

The presence and severity of typical and differential symptoms of LUTI and their impact on the patients’ quality of life were measured using the validated German version of the Acute Cystitis Symptom Score (ACSS) [20]. General health status was additionally assessed using the validated German version of the three-Level EuroQoL 5-Dimensions questionnaire (EQ-5D-3L) [21]. Both questionnaires were completed by the patients.

The Lower Urinary Tract Infection Recurrence Risk (LUTIRE) nomogram was used for risk stratification based on the collected parameters [11]. Data from medical history, clinical examination, and ultrasonography were categorized according to the ORENUC system to assess host-related factors and the complexity of the infection [12].

### 2.4. Urinalyses and Urine Culture

Urine dipstick leucocyte esterase results were classified into five semi-quantitative levels: Negative (up to 10 WBC/µL), Trace (>10 to 25 WBC/µL), Small (1+) (>25 to 50 WBC/µL), Moderate (2+) (>50 to 100 WBC/µL), and Large (3+) (>100 WBC/µL) [22]. CFU counts on urine culture were categorized into six groups: 10^2^, 10^3^, 10^4^, 10^5^, 10^6^, and >10^6^ CFU/mL. Counts up to 10^3^ CFU/mL were considered insignificant. Uropathogens with intrinsic resistance to specific antimicrobials were not tested against those agents, in accordance with the Expert Rules of the European Committee on Antimicrobial Susceptibility Testing (EUCAST) [23,24].

### 2.5. Study Rationale

This post hoc analysis aimed to explore clinical and paraclinical factors associated with recurrent cystitis using prospectively recorded data from the GPIU.COM database. All variables used for the analysis were collected prospectively during the original study [19]. No new variables were constructed after the database lock.

Variables which directly influenced the study definitions were stepwise and manually excluded to avoid circular reasoning, model overfitting, and bias introduced by outcome-related predictors. These were the number of previous symptomatic UTI episodes, prior prophylactic measures and antimicrobial treatments, the LUTIRE recurrence probability, and the “O” and “R” criteria of the ORENUC classification.

### 2.6. Statistical Evaluation

Sample size was calculated using Cochrane’s formula for descriptive and cross-sectional studies [25].

Data were entered into an online database designed for the GPIU.COM study via electronic case report forms (eCRFs). Only fully completed records were eligible for further analysis.

All statistical analyses followed a pre-specified, robust stepwise workflow (Table 1), with sequential exclusion of statistically non-significant variables to obtain the most parsimonious and best-fitting models. The significance threshold was set at *p* < 0.05.

Analysis and graphical representation of the results were performed using R-Studio, integrated with R v.3.5.2 and related packages [26]. Normality of variables was assessed using the Shapiro–Wilk test and Q-Q plots [27]. Homogeneity of variances was examined using Levene’s test [28]. Numerical values were summarized as appropriate measures of central tendency and dispersion (e.g., mean, median, standard deviation, 95% confidence interval (CI), interquartile range). Categorical variables were reported in proportions.

Between-group comparisons included univariate parametric or non-parametric tests, depending on the nature and type of variable, normality of distribution, and frequencies of cases in the groups, including Student’s or Welch’s *t*-test, ANOVA, Wilcoxon rank-sum, Pearson’s chi-squared, and Fisher’s exact proportion tests [29,30,31,32,33,34].

The null hypothesis stated that there were no differences between the groups.

*p*-values for multiple comparisons involving continuous or ordinal variables were adjusted using the Benjamini–Hochberg (BH) method to control the false discovery rate (FDR). Pearson’s product-moment correlation coefficient was used to measure the strength of associations for numerical and interval variables [35]. Association measures were expressed as relative risk (RR) with 95% CIs, calculated to identify variables associated with AC or RC, with a primary focus on risk factors for developing RC. Haldane–Anscombe correction was used for contingency tables containing zero cells to allow valid estimation of RR and CIs [36].

Only significant and non-collinear predictors were entered into univariate logistic regression (LR) models to obtain adjusted odds ratios (OR) with 95% CI and the final model equation.

Predictors that had significant ORs in univariate LR were included in the final multivariate LR model. Model discrimination was assessed using the area under the receiver operating characteristic curve (AUC). Calibration was evaluated using the Hosmer–Lemeshow goodness-of-fit test, and multicollinearity was assessed using the variance inflation factor (VIF).

Internal validation was performed using bootstrap resampling (1000 iterations) to estimate model performance. Discrimination and calibration performance were evaluated via adjusted values of the AUC, calibration slope, and intercept.

## 3. Results

### 3.1. Clinical Assessment

#### 3.1.1. Demographics

A total sample of 157 women was recruited. Fifty-one respondents with incomplete data and those who did not match the study definition of acute LUTI were excluded. Data from 106 patients were categorized into two groups, sporadic acute cystitis (AC) (*n* = 50; 47.2%) and recurrent acute cystitis (RC) (*n* = 56; 52.8%), and included in the final analysis. The groups were comparable for all demographic characteristics (Table 2). The complete set of all tested variables is provided in the Appendix A.

#### 3.1.2. Symptoms

Analysis of the summary scores of the ‘Typical’, ‘Differential’ and ‘Quality of Life’ domains of the ACSS revealed no statistically significant differences between the groups (*p* > 0.4). However, patients with RC more frequently reported a severe impact of their quality of life (QoL), such as a severe impact of symptoms on their daily activities (ACSS), and experienced extreme levels of anxiety and depression (EQ-5D-3L), with a higher tendency to be confined to their bed compared to those with AC (EQ-5D-3L). Results of these comparisons are detailed in Appendix A. Symptoms with significant differences between the groups are presented in Table 3.

#### 3.1.3. History and Risk Factors

While the differentiation into AC and RC based on patients’ information and journal notes was straightforward, assessment according to the LUTIRE and ORENUC classification revealed that patients with AC had also had previous episodes with the same symptoms and had used preventive measures. Some patients with AC also had risk factors for recurrence. In contrast, while risk factors for recurrence were significantly more common among RC patients, some patients in the RC group had no known risk factors for recurrence. Use of classification instruments showed the uncertainty of differentiating AC and RC based on the patients’ history only.

Patients with AC more often reported normal bowel function, and patients with extra-urogenital risk factors like constipation (LUTIRE)(ORENUC-E) had an 83% (15/18) chance of belonging to the RC group (two-sided *p* < 0.05 for both comparisons) (Table 3) (Appendix A). AC patients also more often reported a sense of incomplete bladder emptying on ACSS, but the urologist found no evidence of residual urine (ORENUC-U). Moreover, patients with AC reported more severe flank pain and symptoms of menopause-related complaints (*p* < 0.05) which did not correspond with urological (ORENUC-U) findings or factual menopause status (Table 3). The proportion of patients who presented with multiple abnormalities was significantly higher among patients with RC, indicating that individual risk factors have an additive effect on the overall risk of recurrence (Appendix A).

The proportion of patients with a history of isolation of a Gram-negative uropathogen was significantly higher in the RC group (*p* < 0.001) (Table 3), whereas the proportion of Gram-positive uropathogens did not differ significantly between the groups (*p* = 0.277) (Appendix A).

### 3.2. Urinalysis and Microbiological Findings

#### 3.2.1. Urinalysis

Patients with RC exhibited a significantly higher rate of negative leucocyte esterase tests than AC patients (17.9% vs. 4.0%), as well as a higher frequency of concurrent negative results for leucocyte esterase and nitrite tests (12.5% vs. 2.0%). Conversely, the proportion of patients with the “Moderate (2+)” level on the leucocyte esterase test was significantly higher among patients in AC group (Table 4).

#### 3.2.2. Urine Culture

Out of the 106 urine samples taken on admission from all included patients, 91 (85.8%) yielded a positive culture (CFU ≥ 10^3^/mL). Among these, a single uropathogen was isolated in 67 cultures (73.6%), while two uropathogens were isolated in 24 cultures (26.4%). In total, 119 uropathogens were isolated; 76 (63.9%) were Gram-negative and 33 (27.7%) were Gram-positive. Ten cultures (8.4%) were classified as mixed flora (Appendix A).

*Escherichia coli* was the most frequently isolated primary (first) uropathogen, with a prevalence of 66.7% in the AC and 61.1% in the RC group (Appendix A), and it did not differ significantly between the groups (*p* > 0.05). No significant difference was observed between the groups regarding the proportion of patients with a single uropathogen or a negative urine culture (Appendix A). Patients with RC had a significantly higher rate of multiple uropathogens compared to those with AC (31.3% vs. 6.8%) (*p* < 0.05) (Table 4, Appendix A).

For *E. coli*, the non-susceptibility rate to second-generation cephalosporins was higher in the RC as compared to the AC group. No statistically significant differences between the two groups were observed for other antibiotics tested against *E. coli* uropathogens (Appendix A).

### 3.3. Statistical Analysis

#### 3.3.1. Weighting of Risk Factors Based on Relative Risk (RR)

Based on the incidence of risk factors with statistically significant differences between the groups, we calculated point estimates and 95% confidence intervals (CIs) for RR and present them in a forest plot (Figure 1).

Absence of problems with daily activities according to the ACSS and presence of pyuria were most strongly associated with AC. Conversely, chronic constipation (LUTIRE-nomogram), extra-urogenital risk factors (ORENUC E-category), a negative leucocyte esterase test, a moderate level of red blood cells in urine, and reporting extreme anxiety or depression (EQ-5D-3L) showed the strongest weight for RC (Figure 1, Appendix A).

#### 3.3.2. Evaluation of Predicting Factors for Recurrent Cystitis Based on Odds Ratio (OR)

The clinical, laboratory, and microbiological parameters that showed significant relative risk ratios for either the AC or the RC group were selected for further analyses. Based on the incidence of risk factors with statistically significant differences between groups, we calculated ORs with 95% CI and performed univariate logistic regression. The strongest predictors for AC were pyuria and absence of problems with performing usual activities. The strongest predictors for RC were trace proteinuria and the presence of multiple uropathogens (Figure 2).

#### 3.3.3. Performance and Validity of Predictive Models Based on Multivariate Logistic Regression

Final multivariate logistic regression analysis included variables with the highest ORs for RC. These were chronic constipation, severe impact of symptoms on everyday activities, trace proteinuria, pyuria, and multiple uropathogens isolated from urine. Three alternative models were constructed to identify the strongest set of predictors for RC and AC.

Model I, which included predisposition to constipation (OR 6.73; *p* = 0.012), severe impact of symptoms on everyday activities (OR 6.44; *p* = 0.015), multiple uropathogens (OR 12.53; *p* < 0.001), and trace proteinuria (OR 12.14; *p* = 0.032), showed good discrimination ability (apparent AUC = 0.84; optimism-corrected AUC = 0.82), with a sensitivity of 0.86, specificity of 0.70, and overall accuracy of 78.3% (Figure 3a). Calibration was adequate by the Hosmer–Lemeshow test (*p* = 0.30), with a calibration intercept around zero. The apparent calibration slope was 1.00, and the optimism-corrected slope after 1000 bootstrap resamples was 0.64.

Model II included a history of constipation (OR 5.65; *p* = 0.020), severe impact of symptoms on everyday life (OR 6.47; *p* = 0.012), multiple uropathogens (OR 11.41; *p* = 0.001), and pyuria (OR 0.23; *p* = 0.008) as independent predictors. This model demonstrated robust performance with good discrimination ability (apparent AUC = 0.82; optimism-corrected AUC = 0.80) and balanced calibration (corrected slope = 0.82; intercept = 0). The sensitivity and specificity of this model were 0.82 and 0.72, respectively, and its, overall accuracy was 77.4% (Figure 3b).

Model III, which included a history of constipation (OR 6.50; *p* = 0.011), severe impact of symptoms on everyday activities (OR 5.62; *p* = 0.020), multiple uropathogens (OR 12.12; *p* < 0.001), and trace proteinuria (OR 12.38; *p* = 0.026), demonstrated consistent discrimination (apparent AUC = 0.80; optimism-corrected AUC = 0.79) and good calibration (corrected slope = 0.69; intercept ≈ 0; Hosmer–Lemeshow *p* = 0.87), indicating minimal overfitting and stable internal validity after 1000 bootstrap resamples. Sensitivity, specificity, and overall accuracy were 0.73, 0.80, and 76.4%, respectively (Figure 3c). Further details on all prediction models are provided in the Appendix A.

## 4. Discussion

### 4.1. Main Findings

We assessed patients with acute cystitis by means of ACSS, LUTIRE, and the ORENUC classification and found that 14% of patients considered to have AC reported a similar symptomatic episode within the preceding 6 months and that 74% had used preventive measures against acute cystitis. Patients with AC more often reported severe flank pain, a moderate sense of incomplete bladder emptying, and menopause-related complaints which did not correspond with urological (ORENUC-U) findings or factual menopause status.

Patients with RC more frequently reported a severe impact of symptoms on their daily activities and experienced extreme levels of anxiety and depression, with a higher tendency to be confined to their bed. All patients with extra-urogenital risk factors according to ORENUC and 63% of all patients in the study population with urological abnormalities belonged to the RC group. All patients with more than one ORENUC risk category, except O or R, also had RC.

Patients in the RC group more frequently had a history of Gram-negative uropathogen isolation and more frequently had Gram-positive uropathogens isolated on admission. In cases with multiple uropathogens, patients with RC had a significantly higher prevalence of Enterococcus species as the second uropathogen with higher colony counts compared to those with AC.

Despite the statistically significant differences observed in the study between the AC and RC groups in terms of trace proteinuria and moderate erythrocyturia (Table 4), the interpretation of these findings is limited by the low number of cases. However, the consistently higher prevalence of trace proteinuria and erythrocyturia among patients with recurrent cystitis may reflect persistent urothelial inflammation.

Logistic regression of odds ratios showed that the strongest predictors for AC were pyuria and the absence of problems with performing usual activities. The strongest predictors for RC were trace proteinuria and the presence of multiple uropathogens. Three constructed prediction models showed consistent results, highlighting a history of constipation, severe impact of symptoms on everyday activities, and the presence of multiple uropathogens as the strongest set of independent risk factors for RC, while pyuria was inversely associated with AC.

### 4.2. Methodological Aspects

This is the first time patients with acute cystitis are evaluated for risk factors according to the combined use of ACSS, LUTIRE, and the ORENUC classification. While patients with AC reported severe flank pain, the urologists found no fever, flank tenderness, or dilatation on ultrasonography and hence could rule out pyelonephritis. Likewise, some patients reported a feeling of incomplete bladder emptying, but the urologist found no residual urine. While patients with AC reported symptoms of menopause, the ACSS did not register the hormonal status of menopause, but this information was picked up by LUTIRE. When developing the LUTIRE nomogram, Cai et al. found that hormonal status and constipation had the highest predictive value for recurrence [11]. They also found the type of previously isolated pathogens to be important. This is consistent with our findings. Other important factors identified by Cai et al. were the number of sexual partners and previous treatment of asymptomatic bacteriuria. We did not identify these risk factors in our study. On the other hand, Cai et al. did not identify risk factors in the ORENUC E-category, which in our study were endometriosis, immunosuppression, entero-vesical fistula, multiple sclerosis, and one factor designated as “other”.

Our findings highlight differences and limitations in the reporting of risk factors by patients and urologists. We therefore argue that risk factors with the highest predictive value for RC from ACSS, LUTIRE, and ORENUC should be combined in a comprehensive prognostic evaluation tool. The current differentiation of acute cystitis into AC and RC should be replaced by a new description of acute cystitis with a grading of the risk of recurrence, like that in bladder cancer [37].

The first logistic regression model (Model I) was exploratory and incorporated the five significant predictors identified in the univariate analysis, including chronic constipation, severe impact of symptoms on everyday activity, multiple uropathogens, trace proteinuria, and pyuria. This model enabled the individual contribution and direction of association of each variable to be evaluated. However, as pyuria and trace proteinuria are related urinary inflammatory responses that showed moderate collinearity, two additional models (Models II and III) were constructed. Each of these models excluded one of the potentially associated predictors (trace proteinuria or pyuria) to assess the robustness of the estimates and avoid redundancy in the multivariate framework. Among all models, Model II, which included a history of chronic constipation, severe impact of symptoms on everyday life/activities, multiple uropathogens in urine culture, and pyuria, provided the broadest detection window with the most optimal trade-off between sensitivity and negative predictive value, thereby reducing the likelihood of missing true cases of RC.

Considering the relatively limited number of relevant cases in our study, our observations should be interpreted with caution and confirmed in larger prospective cohorts designed to explore their potential as early markers of recurrence risk. To address this issue, we anticipated the risk of model instability and potential overfitting during multivariate LR, applying internal validation via bootstrap resampling with 1000 iterations for all three LR models. This approach provided optimism-corrected estimates of discrimination and calibration without data splitting [38].

### 4.3. Context and Clinical Impact

Our results underline the importance of a thorough anamnesis and clinical examination with evaluation of extra-urogenital, nephrological, and urological risk factors. The presence of risk factors can help the clinician decide which type of cystitis the patient has and the risk of recurrence. Microbiological results add value to the differentiation between AC and RC. Patients with RC tend to have multiple and more resistant pathogens, which require more broad-spectrum antibiotics as compared with AC. Reporting of incomplete bladder emptying and flank pain among RC patients are arguments for having ultrasonography at hand at the initial visit. A history of constipation should prompt instant guidance on lifestyle habits.

Although patients in the AC group more frequently reported menopausal symptoms and higher symptom burden, our previous studies did not demonstrate a significant association between symptom severity and findings in urinalyses and urine culture [39,40]. These patient complaints are arguments for expanding the ORENUC classifications of UTI to include a mental domain, since menopausal symptoms and symptom burden can lead to depression and anxiety and affect daily activities and work.

Our prediction models are relevant for decision algorithms for diagnosis and treatment of acute episodes of cystitis. Model I included constipation, severe impact of symptoms on everyday activities, trace proteinuria, pyuria, and multiple uropathogens isolated from urine. This model is most valuable when the aim is to avoid missing any RC cases, even at the cost of some over-diagnosis. Model III, which excluded pyuria, is preferable when the priority is to prevent overtreatment and support antibiotic stewardship. Model II, which excluded trace proteinuria, provides a balanced compromise and seems most relevant for routine clinical practice.

### 4.4. Strengths and Limitations

Key strengths of our study are its prospective design, inclusion of symptomatic patients only, clinical examination by a urologist at baseline, availability of urine culture and antimicrobial susceptibility data from nearly all patients, and a robust stepwise statistical evaluation. The use of standardized tools such as ACSS and structured categorization systems (LUTIRE and ORENUC), as well as graded laboratory assessments, ensured comparability between patient groups.

A limitation of this study is that some of the predictors conceptually overlapped with diagnostic criteria, posing a potential risk of collinearity. We therefore excluded the most collinear factors from the final analysis. Classic determinants such as sexual activity, diabetes mellitus, menopausal status, and recent antibiotic exposure were not included in the regression models. This might represent residual confounding and limit contextualization of our findings.

The modest sample size could also have limited external generalizability, so we compensated for this with rigorous stepwise statistical methodology and appropriate statistical corrections. Specifically, bootstrap resampling with 1000 iterations was applied to all multivariate logistic models to obtain optimism-corrected estimates of discrimination and calibration without sacrificing data for split-sample validation.

Although we classified CFU counts ≤10^2^/mL as insignificant, the use of a threshold of 10^3^ CFU/mL may still have led to the inclusion of patients who would be considered bacteriuria-negative in other studies. Finally, as our study was limited to Germany and Switzerland, the findings may not be fully generalizable to other geographic regions with different microbial patterns of non-susceptibility.

### 4.5. Future Research

Reporting of severe flank pain and moderate findings on leucocyte esterase test in AC, as opposed to a negative leucocyte esterase test, trace proteinuria, and moderate red blood cells in RC, warrants further immunological characterization of the two clinical entities. More frequent findings of multiple uropathogens, more often in RC than in AC, indicate different pathogenetic mechanisms. The above features may generate hypotheses of a different innate immunology in patients with AC and RC to be addressed in future studies with more specific tests [16].

Future research should address the role of the intestinal microbiome in constipation-induced RC and explore why the bladder mucosa is more susceptible to infection in cases of poor bladder emptying. We need large prospective studies to assess the prognostic weight of individual factors and the effect of eliminating separate risk factors on recurrence rates. Our models for the identification of recurrent episodes of acute cystitis should be expanded by means of AI into larger decision algorithms to guide treatment and prevention policies.

## 5. Conclusions

Although both sporadic and recurrent cystitis present as acute infection of the urinary bladder, there are significant differences in symptoms, findings on urinalysis and urine culture, and risk factors according to ACSS, LUTIRE, and ORENUC classifications. The most common risk factors among patients with RC are constipation and urological abnormalities. Use of risk factor classifications will ensure a comprehensive patient evaluation and help differentiate between AC and RC. The strongest predictors for sporadic cystitis are pyuria and the absence of problems with performing usual activities. The strongest predictors for recurrent cystitis are trace proteinuria and the presence of multiple uropathogens. Different findings on urinalysis and culture indicate different host reactions and pathogenetic mechanisms in the two clinical conditions and warrant further research.

## Figures and Tables

**Figure 1 diagnostics-15-02885-f001:**
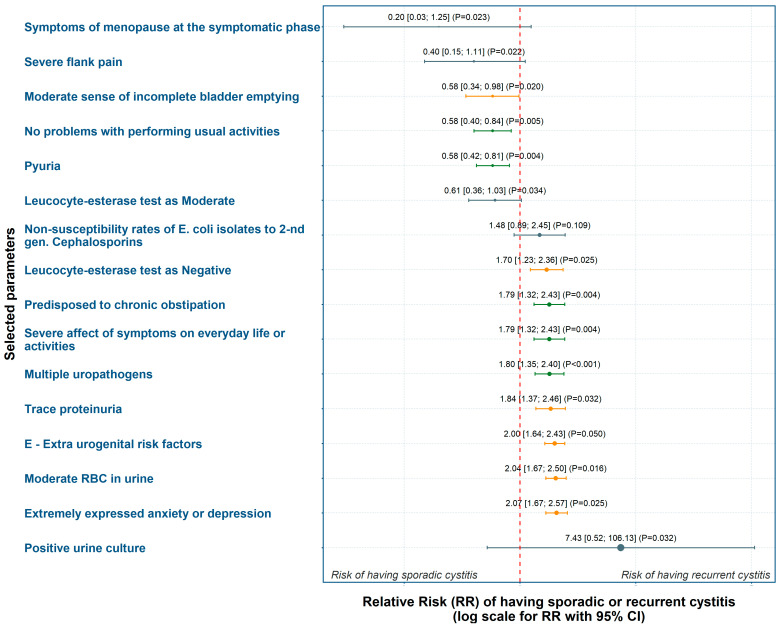
Forest plot of RR for having sporadic or recurrent cystitis. Values to the left of the red dashed line (RR < 1.0) indicate stronger association with acute cystitis, whereas those to the right (RR > 1.0) indicate association with recurrent cystitis. Color code: green—*p* ≤ 0.01; orange—0.01 < *p* ≤ 0.05; gray—*p* > 0.05. Error bars represent 95% confidence intervals.

**Figure 2 diagnostics-15-02885-f002:**
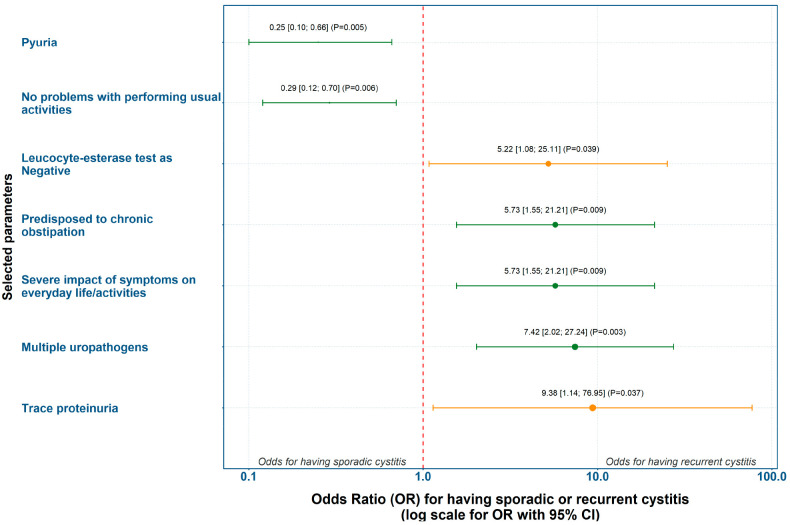
Forest plot of OR for having sporadic or recurrent cystitis. Color code: green—*p* ≤ 0.01; orange—0.01 < *p* ≤ 0.05; gray—*p* > 0.05.

**Figure 3 diagnostics-15-02885-f003:**
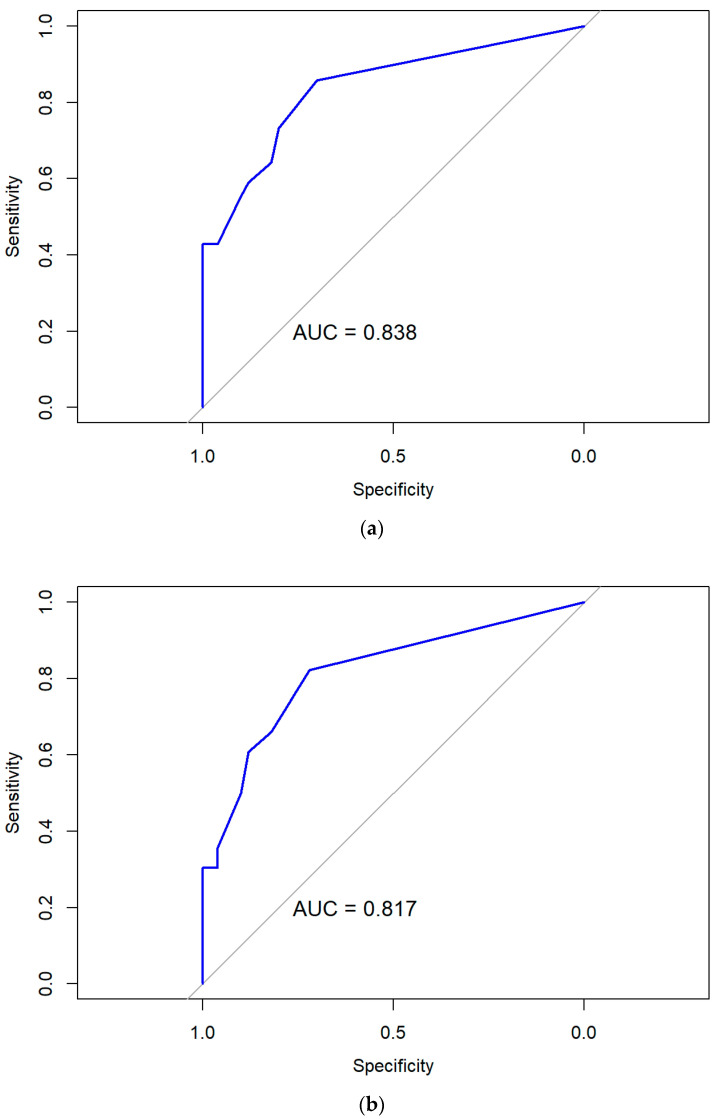
ROC-curves of the final logistic regression models for: (**a**) Model I; (**b**) Model II; and (**c**) Model III.

**Table 1 diagnostics-15-02885-t001:** Stepwise statistical analysis of study data.

Step	Statistical Method	Purpose	Input Variables	Outcome/Value
1	Distribution and variance testing	Assess normality and homogeneity of variance	All continuous variables	Suitability of variables for parametric vs. non-parametric tests
2	Descriptive statistics	Characterize the study population and variables	Demographics, symptoms, lab data, ACSS, ORENUC, LUTIRE	Central tendency and distribution
3	Between-group comparative analysis	Identify differences between AC and RC	Each candidate variable	Variables with *p* < 0.05 are selected for the next step
4	Correlation and redundancy analysis, manual exclusion	Remove redundant or collinear predictors	Significant variables from Step 3	Reduced set of potential predictors
5	Association measures	Assess direction and magnitude of association	Selected set of potential predictors	Relative risk (RR) with 95% CI
6	Logistic regression (binomial, stepwise)	Build the final predictive model	Independent predictors from Steps 4 and 5	Odds ratios with 95% CI, model equation
7	Assessment of model performance and internal validation	Evaluate discrimination, calibration, and multicollinearity; verify stability of predictive performance	Final model output	AUC > 0.8, *p* > 0.05 (Hosmer–Lemeshow), VIF < 5, bootstrap resampling

**Table 2 diagnostics-15-02885-t002:** Patient demographics.

Demographic Characteristics	Total	Sporadic Acute Cystitis (AC)	Recurrent Cystitis (RC)	*p*-Value (Significance)
Number of patients, *n* (%)	106 (100.0)	50 (47.2)	56 (52.8)	n.a.
Age, median (IQR)	36.5 (26.0–58.5)	36.0 (27.2–51.0)	37.5 (24.0–66.5)	0.219 (ns) *
Weight in kg, median (IQR)	66.0 (57.0–74.8)	63.0 (56.0–70.0)	67.5 (60.0–75.0)	0.191 (ns) *
Height in m, median (IQR)	1.6 (1.6–1.7)	1.6 (1.6–1.7)	1.7 (1.6–1.7)	0.054 (ns) *
Body mass index, median (IQR)	23.4 (21.3–27.5)	23.0 (21.3–27.2)	23.4 (21.4–27.7)	0.477 (ns) *
Pregnancy, *n* (%)	2 (1.9)	1 (2.0)	1 (1.8)	1.000 (ns) ‡

Note: n.a.—not applicable; IQR—interquartile range; *n*—number; ns—non-significant difference; *—Student’s/Welch’s t-test; ‡—Fisher’s exact proportion test. For details, see Appendix A.

**Table 3 diagnostics-15-02885-t003:** Data from the medical history of study respondents (includes only variables showing statistically significant differences between the groups).

Data from the Medical History		Total	Sporadic Acute Cystitis (AC, *n* = 50)	Recurrent Cystitis (RC, *n* = 56)	*p*-Value (Significance)
**Anamnestic Data**	At least one prior symptomatic episode of UTIs in the past 6 months, *n* (%)	38 (35.8)	7 (14.0)	31 (55.4)	<0.001 (****) †
	At least one prior symptomatic episode of UTIs in the past 12 months, *n* (%)	64 (60.4)	8 (16.0)	56 (100.0)	<0.001 (****) †
	Number of prior symptomatic episodes UTIs in the past 6 months, median (IQR)	2.0 (1.0–5.0)	1.0 (0.0–1.0)	3.0 (2.0–5.0)	<0.001 (****) ‡
	Number of prior symptomatic episodes UTIs in the past 12 months, median (IQR)	3.0 (0.0–12.0)	0.0 (0.0–0.0)	11.0 (5.0–40.0)	<0.001 (****) ‡
	No prophylactic measure in the past 12 months, *n* (%)	55 (51.9)	37 (74.0)	18 (32.1)	<0.001 (****) †
	Multiple prophylactic measures in the past 12 months, *n* (%)	43 (40.6)	10 (20.0)	33 (58.9)	<0.001 (***) †
**ACSS**	Symptoms of menopause, *n* (%)	9 (8.5)	8 (16.0)	1 (1.8)	0.012 (*) ‡
	Moderate sense of incomplete bladder emptying, *n* (%)	29 (27.4)	19 (38.0)	10 (17.9)	0.035 (*) †
	Severe flank pain, *n* (%)	13 (12.3)	10 (20.0)	3 (5.4)	0.035 (*) ‡
	Severe impact of UTI symptoms on everyday life/activities, (*n*%)	18 (17.0)	3 (6.0)	15 (26.8)	0.005 (**) ‡
**EQ-5D-3L**	No problems with performing usual activities, *n* (%)	59 (55.7)	35 (70.0)	24 (42.9)	0.009 (**) †
	Extremely anxious or depressed, *n* (%)	7 (6.6)	0 (0.0)	7 (12.5)	0.014 (*) ‡
**LUTIRE**	Predisposed to chronic constipation, according to the LUTIRE nomogram, *n* (%)	18 (17.0)	3 (6.0)	15 (26.8)	0.005 (**) ‡
	Known Gram-negative uropathogen isolated at the last acute episode, according to the LUTIRE nomogram, *n* (%)	25 (23.6)	4 (8.0)	21 (37.5)	<0.001 (***) ‡
	No known uropathogen in the past, according to the LUTIRE nomogram, *n* (%)	73 (68.9)	44 (88.0)	29 (51.8)	<0.001 (***) †
	Up to 2 acute episodes per year, according to the LUTIRE nomogram, *n* (%)	52 (49.1)	50 (100.0)	2 (3.6)	<0.001 (****) ‡
	Three or more acute episodes per year, according to the LUTIRE nomogram, *n* (%)	54 (50.9)	0 (0.0)	54 (96.4)	<0.001 (****) ‡
	Probability of recurrence according to the LUTIRE nomogram, median (IQR)	0.30 (0.20–0.40)	0.20 (0.20–0.30)	0.40 (0.30–0.50)	<0.001 (****) •
**ORENUC**	O—No known risk factor, *n* (%)	56 (52.8)	37 (74.0)	19 (33.9)	<0.001 (****) †
	R—Risk factors for recurrent UTIs, but no risk of more severe outcome, *n* (%)	32 (30.2)	9 (18.0)	23 (41.1)	0.018 (*) †
	E—Extra-urogenital risk factors with risk of more severe outcome, *n* (%)	6 (5.7)	0 (0.0)	6 (10.7)	0.028 (*)

Note: * *p* < 0.05; ** *p* < 0.01; *** *p* < 0.001; **** *p* < 0.0001; *n*—number; IQR—interquartile range; †—Pearson’s chi-square proportion test; ‡—Fisher’s exact proportion test; •—Wilcoxon rank-sum test. For details, see Appendix A.

**Table 4 diagnostics-15-02885-t004:** Results of urinalysis and urine microbiology showing statistically significant differences between the groups.

Results of Urine Tests at Baseline Visit	Total	Sporadic Acute Cystitis (AC, *n* = 50)	Recurrent Cystitis (RC, *n* = 56)	*p*-Value (Significance)
Negative leucocyte esterase test, *n* (%)	12 (11.3)	2 (4.0)	10 (17.9)	0.032 (*) ‡
Moderate leucocyte esterase test, *n* (%)	28 (26.4)	18 (36.0)	10 (17.9)	0.001 (**) †
Pyuria *, *n* (%)	77 (72.6)	43 (86.0)	34 (60.7)	0.007 (*) †
Moderate erythrocyturia, *n* (%)	8 (7.5)	0 (0.0)	8 (14.3)	0.006 (**) ‡
Trace proteinuria, *n* (%)	10 (9.4)	1 (2.0)	9 (16.1)	0.018 (*) ‡
Positive urine culture (CFU ≥ 10^3^/mL), *n* (%)	87 (82.1)	33 (66.0)	54 (96.4)	<0.001 (***) †
Multiple uropathogens, *n* (%)	24 (22.6)	3 (6.0)	21 (37.5)	<0.001 (***) ‡
Non-susceptibility rates of *E. coli* isolates to 2-nd gen. Cephalosporins, *n* (%)	28 (28.9)	9 (17.6)	19 (41.3)	0.019 (*) †

Note: * Pyuria is defined as leucocyte esterase test results as Moderate (2+) and Large (3+); * *p* < 0.05; ** *p* < 0.01; *** *p* < 0.001; *n*—number; †—Pearson’s chi-square proportion test; ‡—Fisher’s exact proportion test. For details, see Appendix A.

## Data Availability

The data presented in this study are available on request from the corresponding author.

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
