# Peer review of "Identification of Risk Factors in Patients with Recurrent Cystitis May Improve Individualized Management"

_diagnostics, 2025, doi:10.3390/diagnostics15222885_

Round 1
Reviewer 1 Report
Comments and Suggestions for Authors
The manuscript addresses a clinically relevant issue — the differentiation between sporadic and recurrent cystitis and its implications for individualized management. The integration of the LUTIRE and ORENUC classifications adds potential clinical value, but the work is limited by a small and geographically restricted cohort, limited power, and possible overfitting in multivariate analyses. Some methodological details, particularly regarding the post-hoc nature of the analysis and the criteria used for variable selection, should be clarified.
Overall, the study contributes useful descriptive information, but several conclusions (e.g., immunological interpretations) extend beyond the presented data. I would recommend major revision before potential acceptance, contingent upon stronger methodological transparency and tempered interpretation of findings.
1. Clarify the rationale and boundaries of the post-hoc design.
While the manuscript defines this as a post-hoc analysis of the GPIU.COM dataset, it would benefit from a clearer explanation of which variables were prospectively collected and which were selected retrospectively for this secondary analysis. This clarification would strengthen the methodological transparency, since without it, it is difficult to assess whether the observed associations were pre-specified or data-driven.
2. Provide more information on sample selection and representativeness.
Only 106 women were finally analyzed despite the long inclusion period and the study being described as multinational. It would help the reader to understand how many were screened, excluded, or lost to follow-up. This is important because the small final cohort and its restriction to Germany and Switzerland may limit generalizability and introduce selection bias.
3. Reconsider the conceptual separation between AC and RC in the modeling.
Some of the variables used to distinguish between acute and recurrent cystitis appear to overlap with the outcome definition itself (e.g., frequency of prior episodes). This overlap can introduce circular reasoning and artificially inflate associations. Clarifying how the dependent and independent variables were separated would reinforce the validity of the risk-factor interpretation.
4. Explain more precisely how LUTIRE and ORENUC classifications were applied.
The text refers to both systems, yet it is unclear whether they were used independently, combined, or hierarchically. Given that LUTIRE was designed for prospective recurrence prediction, its application in a retrospective, cross-sectional context requires justification. Without that clarification, the reader may question the appropriateness of the combined analytical framework.
5. Re-examine the antimicrobial susceptibility findings for plausibility.
The statement that overall non-susceptibility rates were higher in the AC group seems counterintuitive, as recurrent cases typically display greater exposure to antimicrobials and, consequently, higher resistance rates. This pattern deserves verification to rule out data misinterpretation or possible mislabeling between groups.
6. Address the potential overfitting in multivariate models.
Three logistic models were constructed with several correlated predictors despite a limited number of events per variable. Reporting internal validation (e.g., cross-validation or penalized regression) would enhance confidence in model robustness. As presented, the relatively high AUCs may reflect overfitting rather than genuine predictive performance.
7. Temper the interpretation of statistically significant but clinically minor findings.
Variables such as “trace proteinuria” or “moderate erythrocyturia” reached statistical significance, yet their clinical impact remains uncertain. Discussing the magnitude and potential biological relevance of these effects—beyond p-values—would help avoid overstatement of their importance.
8. Consider adjustment for additional confounders known to influence recurrence.
Classic determinants such as sexual activity, diabetes, menopausal status, and recent antibiotic exposure were not included in the regression models. Acknowledging this limitation, or clarifying their absence from the dataset, would demonstrate awareness of residual confounding and contextualize the findings more accurately.
9. Include some form of model validation or calibration analysis.
Although discrimination indices (AUCs) are reported, there is no internal or external validation to test model stability. Implementing bootstrap or split-sample validation would substantiate that these predictive models retain accuracy beyond the original sample.
10. Avoid speculative physiological interpretations not supported by the data.
The conclusion suggesting that patients with acute cystitis may have “stronger innate immunity” is intriguing but not empirically evaluated here. It would be more appropriate to frame this as a hypothesis for future research rather than as an inference from the current findings.
Author Response
- Clarify the rationale and boundaries of the post-hoc design.
While the manuscript defines this as a post-hoc analysis of the GPIU.COM dataset, it would benefit from a clearer explanation of which variables were prospectively collected and which were selected retrospectively for this secondary analysis. This clarification would strengthen the methodological transparency, since without it, it is difficult to assess whether the observed associations were pre-specified or data-driven.
Respond: We appreciate this valuable comment.
To improve methodological transparency, we have added a new subsection (“Study rationale”) to the “Materials and Methods”, to clarify the rationale and boundaries of the post-hoc design.
This new subsection explicitly states that all variables used in the present analysis were prospectively collected during the original GPIU.COM study, and that no new variables were constructed after the database lock. Furthermore, we also emphasize that all variables that directly influenced the study’s outcome definition were stepwise and manually excluded to avoid circular reasoning, model overfitting, and bias related to outcome-dependent predictors.
- Provide more information on sample selection and representativeness.
Only 106 women were finally analyzed despite the long inclusion period and the study being described as multinational. It would help the reader to understand how many were screened, excluded, or lost to follow-up. This is important because the small final cohort and its restriction to Germany and Switzerland may limit generalizability and introduce selection bias.
Respond: We thank the reviewer for this valuable comment.
To improve clarity and transparency regarding cohort selection, we have added a short sentence to the “Results” section explaining the total number of screened participants, the number and reasons for exclusions, and the final number included in the analysis. We hope that this addition clarifies the exclusion. Page 5, line 169.
- Reconsider the conceptual separation between AC and RC in the modeling.
Some of the variables used to distinguish between acute and recurrent cystitis appear to overlap with the outcome definition itself (e.g., frequency of prior episodes). This overlap can introduce circular reasoning and artificially inflate associations. Clarifying how the dependent and independent variables were separated would reinforce the validity of the risk-factor interpretation.
Respond: This overlap is very well spotted. Indeed, variables directly influencing the definition of recurrence were excluded from all regression analyses to prevent circular reasoning and overestimation of associations.
This clarification was already present in the “Statistical Analysis” subsection, but for better logical flow and visibility, we have moved and expanded it into the new subsection “Study rationale” within “Materials and Methods”. Kindly see our response to point 1 above.
- Explain more precisely how LUTIRE and ORENUC classifications were applied.
The text refers to both systems, yet it is unclear whether they were used independently, combined, or hierarchically. Given that LUTIRE was designed for prospective recurrence prediction, its application in a retrospective, cross-sectional context requires justification. Without that clarification, the reader may question the appropriateness of the combined analytical framework.
Respond: We are grateful for another valuable comment.
The LUTIRE and ORENUC classifications were applied independently, each serving a distinct analytical purpose. The LUTIRE nomogram was not used for prediction, but for descriptive risk profiling, while the ORENUC system was employed to define host-related factors and infection complexity to better stratify patients according to their risk of developing complicated UTIs. We have revised and clarified the corresponding paragraphs in the “Materials and Methods” to ensure methodological transparency and avoid any possible misinterpretations. The new sentences are on page 3, lines 110 to 112.
- Re-examine the antimicrobial susceptibility findings for plausibility.
The statement that overall non-susceptibility rates were higher in the AC group seems counterintuitive, as recurrent cases typically display greater exposure to antimicrobials and, consequently, higher resistance rates. This pattern deserves verification to rule out data misinterpretation or possible mislabeling between groups.
Respond: Thank you again for your valuable response. You are absolutely right that the overall non-susceptibility rates were higher in the AC than in the RC group does not make sense. We analysed again the data we have and realised that in the two groups of patients, not always the same antibiotics were tested. Therefore, the overall data cannot be compared. As a consequence, we will take out the overall results and present only the resistance data of E. coli against well-defined antibiotics.
- Address the potential overfitting in multivariate models.
Three logistic models were constructed with several correlated predictors despite a limited number of events per variable. Reporting internal validation (e.g., cross-validation or penalized regression) would enhance confidence in model robustness. As presented, the relatively high AUCs may reflect overfitting rather than genuine predictive performance.
Respond: We thank the Reviewer for this valuable remark.
To mitigate this risk, we performed internal validation using bootstrap resampling with 1,000 iterations, which yielded optimism-adjusted estimates of area under the curve (AUC), calibration slope, and intercept for each multivariate model.
This extra procedure is mentioned in the Materials and Methods section, page 4, lines 160 to 162 and the adjusted performance metrics are added to the Results section, page 10, lines 261 to 277. Also, a paragraph has been added to the Discussion section, page 13, lines 334 to 338. An additional appropriate reference 38 was also added.
- Temper the interpretation of statistically significant but clinically minor findings.
Variables such as “trace proteinuria” or “moderate erythrocyturia” reached statistical significance, yet their clinical impact remains uncertain. Discussing the magnitude and potential biological relevance of these effects—beyond p-values—would help avoid overstatement of their importance.
Respond: We thank the reviewer for this thoughtful and constructive comment.
We agree that the statistical significance of certain urinalysis findings, such as trace proteinuria and moderate erythrocyturia, should not be overstated given their limited clinical magnitude and the relatively small number of cases.
To address this, we have added a paragraph in the Discussion section to clarify the interpretation of these findings and to acknowledge the limitations related to sample size, page 12, lines 296 to 299.
- Consider adjustment for additional confounders known to influence recurrence.
Classic determinants such as sexual activity, diabetes, menopausal status, and recent antibiotic exposure were not included in the regression models. Acknowledging this limitation, or clarifying their absence from the dataset, would demonstrate awareness of residual confounding and contextualize the findings more accurately.
Respond: These variables were excluded during the stepwise statistical workflow due to their insignificance and/or relation to the study definitions. Factors selected for the final steps of the analysis and modelling are given in the Supplementary Table 6.
- Include some form of model validation or calibration analysis.
Although discrimination indices (AUCs) are reported, there is no internal or external validation to test model stability. Implementing bootstrap or split-sample validation would substantiate that these predictive models retain accuracy beyond the original sample.
Respond: We thank the Reviewer for this important methodological comment.
As explained in our response to Comment 6, we have now performed internal validation using bootstrap resampling with 1,000 iterations for all multivariate logistic regression models.
This procedure provided optimism-corrected estimates of model discrimination (AUC) and calibration (slope and intercept), thus offering a robust assessment of model stability and predictive reliability.
The corresponding methodological description and the results of this bootstrap validation have been incorporated into the Materials and Methods, Results, and Discussion sections of the revised manuscript as described in our response to comment number 6 above.
- Avoid speculative physiological interpretations not supported by the data.
The conclusion suggesting that patients with acute cystitis may have “stronger innate immunity” is intriguing but not empirically evaluated here. It would be more appropriate to frame this as a hypothesis for future research rather than as an inference from the current findings.
Respond: We agree with the reviewer`s comment and have taken out all speculative physiological interpretations from the manuscript. The remaining sentence on pathophysiology in the subsection on main findings now reads: “However, the consistently higher prevalence of trace proteinuria and erythrocyturia among patients with recurrent cystitis may reflect persistent urothelial inflammation”, page 12 lines 298 to 299.
We would like to sincerely thank the Reviewer for the exceptionally thorough and constructive feedback.
The detailed and thoughtful comments have significantly contributed to improving the scientific quality, methodological clarity, and interpretive balance of our manuscript.
We highly appreciate the time and expertise invested in the review process and believe that the revised version has greatly benefited from your valuable input.
Reviewer 2 Report
Comments and Suggestions for Authors
This article provides a good review on the identification of risk factors for recurrent cystitis. It is a good review because recurrent cystitis is very common in clinical practice and difficult to be completely cured. The data in this article is detailed, the logic is appropriate, and the structure is reasonable. It is recommended for publication.
Author Response
This article provides a good review on the identification of risk factors for recurrent cystitis. It is a good review because recurrent cystitis is very common in clinical practice and difficult to be completely cured. The data in this article is detailed, the logic is appropriate, and the structure is reasonable. It is recommended for publication.
Respond: We sincerely thank the Reviewer for the kind and encouraging feedback.
We are truly grateful for your positive evaluation of our work and your recognition of its clinical relevance and methodological clarity. Your words mean a lot to us, and we highly appreciate your recommendation for publication.
Reviewer 3 Report
Comments and Suggestions for Authors
The present article search for clinical and paraclinical risk factors associated with recurrent cystitis.The article is a model of well presentation,with clear research question and thoroughly statistical analysis.Results are presented in details,discussions are related to results and comparative with the previous studies..Minor comments:
-row 94 >102CFU/ml please correct
-row 66 "some interventions" I would specify what interventions failed to show efficacy
-you have utilised in text "trace proteinuria" and "proteinuria" interchangeable.I would recommend only one term,namely trace proteinuria.
- It would be helpful to provide more comments in the discussion part about the relationship that could exist between proteinuria and RC.The same for the inversely association between pyuria and RC.
Author Response
-row 94 >102CFU/ml please correct
Respond: This typo has been corrected
-row 66 "some interventions" I would specify what interventions failed to show efficacy
Respond: This has also been corrected, and the sentence now reads “ some interventions (e.g. D-mannose)” page 2, line 68
-you have utilised in text "trace proteinuria" and "proteinuria" interchangeable. I would recommend only one term,namely trace proteinuria.
Respond: A single term is now used throughout the text.
- It would be helpful to provide more comments in the discussion part about the relationship that could exist between proteinuria and RC.The same for the inversely association between pyuria and RC.
Respond: The relationship that could exist between proteinuria and RC has now been addressed in the subsection on main findings. Kindy see our response to reviewer 1, point 10 above.
In line with the criticism from reviewer 1, our comments on immunological mechanisms now reads: “The above features may generate hypotheses of a different innate immunology in patients with AC and RC to be addressed in future studies with more specific tests”.
We would like to thank the Reviewer sincerely for the careful reading of our manuscript and for these valuable and constructive remarks.
All suggested corrections have been implemented, and we are confident that the revised version has significantly benefited from your insightful comments and attention to detail.